# PHASE COLLAPSE IN NEURAL NETWORKS

**Florentin Guth, John Zarka**
DI, ENS, CNRS, PSL University, Paris, France
{florentin.guth,john.zarka}@ens.fr

**Stéphane Mallat**
Collège de France, Paris, France
Flatiron Institute, New York, USA

## ABSTRACT

Deep convolutional classifiers linearly separate image classes and improve accuracy as depth increases. They progressively reduce the spatial dimension whereas the number of channels grows with depth. Spatial variability is therefore transformed into variability along channels. A fundamental challenge is to understand the role of non-linearities together with convolutional filters in this transformation. ReLUs with biases are often interpreted as thresholding operators that improve discrimination through sparsity. This paper demonstrates that it is a different mechanism called *phase collapse* which eliminates spatial variability while linearly separating classes. We show that collapsing the phases of complex wavelet coefficients is sufficient to reach the classification accuracy of ResNets of similar depths. However, replacing the phase collapses with thresholding operators that enforce sparsity considerably degrades the performance. We explain these numerical results by showing that the iteration of phase collapses progressively improves separation of classes, as opposed to thresholding non-linearities.

## 1 INTRODUCTION

CNN image classifiers progressively eliminate spatial variables through iterated filterings and subsamplings, while linear classification accuracy improves as depth increases (Oyallon, 2017). It has also been numerically observed that CNNs concentrate training samples of each class in small separated regions of a progressively lower-dimensional space. It can ultimately produce a *neural collapse* (Papyan et al., 2020), where all training samples of each class are mapped to a single point. In this case, the elimination of spatial variables comes with a collapse of within-class variability and perfect linear separability. This increase in linear classification accuracy is obtained in standard CNN architectures like ResNets from the iteration of linear convolutional operators and ReLUs with biases.

A difficulty in understanding the underlying mathematics comes from the flexibility of ReLUs. Indeed, a linear combination of biased ReLUs can approximate any non-linearity. Many papers interpret iterations on ReLUs and linear operators as sparse code computations (Sun et al., 2018; Sulam et al., 2018; 2019; Mahdizadehaghdam et al., 2019; Zarka et al., 2020; 2021). We show that it is a different mechanism, called *phase collapse*, which underlies the increase in classification accuracy of these architectures. A phase collapse is the elimination of phases of complex-valued wavelet coefficients with a modulus, which we show to concentrate spatial variability. This is demonstrated by introducing a structured convolutional neural network with wavelet filters and no biases.

Section 2 introduces and explains phase collapses. Complex-valued representations are used because they reveal the mathematics of spatial variability. Indeed, translations are diagonalized in the Fourier basis, where they become a complex phase shift. Invariants to translations are computed with a modulus, which collapses the phases of this complex representation. Section 2 explains how this can improve linear classification. Phase collapses can also be calculated with ReLUs and real filters. A CNN with complex-valued filters is indeed just a particular instance of a real-valued CNN, whose channels are paired together to define complex numbers.

Section 3 demonstrates the role of phase collapse in deep classification architectures. It introduces a Learned Scattering network with phase collapses. This network applies a learned $1 \times 1$ convolutional complex operator $P_j$ on each layer $x_j$, followed by a phase collapse, which is obtained with a complex wavelet filtering operator $W$ and a modulus:

$$x_{j+1} = |W P_j x_j|. \tag{1}$$

It does not use any bias. This network architecture is illustrated in Figure 1. With the addition of skip-connections, we show that this phase collapse network reaches ResNet accuracy on ImageNet and CIFAR-10.

Section 4 compares phase collapses with other non-linearities such as thresholdings or more general amplitude reduction operators. Such non-linearities can enforce sparsity but do not modify the phase. We show that the accuracy of a Learned Scattering network is considerably reduced when the phase collapse modulus is replaced by soft-thresholdings with learned biases. This is also true of more general phase-preserving non-linearities and architectures.

Section 5 explains the performance of iterated phase collapses by showing that each phase collapse progressively improves linear discriminability. On the opposite, the improvements in classification accuracy of successive sparse code computations are shown to quickly saturate.

The main contribution of this paper is a demonstration that the classification accuracy of deep neural networks mostly relies on phase collapses, which are sufficient to linearly separate the different classes on natural image databases. This is captured by the Learned Scattering architecture which reaches ResNet-18 accuracy on ImageNet and CIFAR-10. We also show that phase collapses are necessary to reach this accuracy, by demonstrating numerically and theoretically that iterating phase-preserving non-linearities leads to a significantly worse performance.

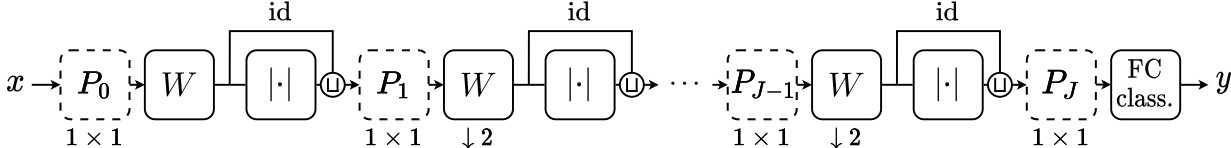

Figure 1: Architecture of a Learned Scattering network with phase collapses. It has $J + 1$ layers with $J = 11$ for ImageNet and $J = 8$ for CIFAR-10. Each layer is computed with a $1 \times 1$ convolutional operator $P_j$ which linearly combines channels. It is followed by a phase collapse, computed with a spatial convolutional filtering with a complex wavelet $W$ and a complex modulus $|\cdot|$. A layer of depth $j$ corresponds to a scale $2^{j/2}$ and a subsampling by 2 is applied every two layers, after $W$. A skip-connection concatenates the outputs of $WP_j$ and $|WP_j|$. A final $1 \times 1$ $P_J$ reduces the dimension before a linear classifier.

## 2 ELIMINATING SPATIAL VARIABILITY WITH PHASE COLLAPSES

Deep convolutional classifiers achieve linear separation of image classes. We show that linear classification on raw images has a poor accuracy because image classes are invariant to local translations. This geometric within-class variability takes the form of random phase fluctuations, and as a result all classes have a zero mean. To improve classification accuracy, non-linear operators must separate class means, which therefore requires to collapse these phase fluctuations.

**Translations and phase shifts** Translations capture the spatial topology of the grid on which the image is defined. These translations are transformed into phase shifts by a Fourier transform. We prove that this remains approximately valid for images convolved with appropriate complex filters.

Let $x$ be an image indexed by $u \in \mathbb{Z}^2$. We write $x_\tau(u) = x(u - \tau)$ the translation of $x$ by $\tau$. It is diagonalized by the Fourier transform $\widehat{x}(\omega) = \sum_u x(u) e^{-i\omega \cdot u}$, which creates a phase shift:

$$\widehat{x_\tau}(\omega) = e^{-i\omega \cdot \tau} \widehat{x}(\omega). \tag{2}$$

This diagonalization explains the need to introduce complex numbers to analyze the mathematical properties of geometric within-class variabilities. Computations can however be carried with real numbers, as we will show.

A Fourier transform is computed by filtering $x$ with complex exponentials $e^{i\omega \cdot u}$. One may replace these by complex wavelet filters $\psi$ that are localized in space and in the Fourier domain. The following theorem proves that small translations can still be approximated by a phase shift in this case. We denote by $*$ the convolution of images.

**Theorem 1.** *Let $\psi\colon \mathbb{Z}^2 \to \mathbb{C}$ be a filter with $\|\psi\|_2 = 1$, whose center frequency $\xi$ and bandwidth $\sigma$ are defined by:*

$$\xi = \frac{1}{(2\pi)^2} \int_{[-\pi,\pi]^2} \omega\, |\widehat{\psi}(\omega)|^2 \,\mathrm{d}\omega \ \ and \ \ \sigma^2 = \frac{1}{(2\pi)^2} \int_{[-\pi,\pi]^2} |\omega - \xi|^2 |\widehat{\psi}(\omega)|^2 \,\mathrm{d}\omega.$$

*Then, for any $\tau \in \mathbb{Z}^2$,*

$$\|x_\tau * \psi - e^{-i\xi\cdot\tau}(x*\psi)\|_\infty \le \sigma\,|\tau|\,\|x\|_2. \tag{3}$$

The proof is in Appendix C. This theorem proves that if $|\tau| \ll 1/\sigma$ then $x_\tau * \psi \approx e^{-i\xi\cdot\tau} x * \psi$. In this case, a translation by $\tau$ produces a phase shift by $\xi \cdot \tau$.

**Phase collapse and stationarity**   We define a *phase collapse* as the elimination of the phase created by a spatial filtering with a complex wavelet $\psi$. We now show that phase collapses improve linear classification of classes that are invariant to global or local translations.

The training images corresponding to the class label $y$ may be represented as the realizations of a random vector $X_y$. To achieve linear separation, it is sufficient that class means $\mathbb{E}[X_y]$ are separated and within-class variances around these means are small enough (Hastie et al., 2009). The goal of classification is to find a representation of the input images in which these properties hold.

To simplify the analysis, we consider the particular case where each class $y$ is invariant to translations. More precisely, each random vector $X_y$ is stationary, which means that its probability distribution is invariant to translations. Equation (2) then implies that the phases of Fourier coefficients of $X_y$ are uniformly distributed in $[0, 2\pi]$, leading to $\mathbb{E}[\widehat{X}_y(\omega)] = 0$ for $\omega \neq 0$. The class means $\mathbb{E}[X_y]$ are thus constant images whose pixel values are all equal to $\mathbb{E}[\widehat{X}_y(0)]$. A linear classifier can then only rely on the average colors of the classes, which are often equal in practice. It thus cannot discriminate such translation-invariant classes.

Eliminating uniform phase fluctuations of non-zero frequencies is thus necessary to create separated class means, which can be achieved with the modulus of the Fourier transform. It is a translation-invariant representation: $|\widehat{x}_\tau| = |\widehat{x}|$. This improves linear discriminability of stationary classes, because $\mathbb{E}[|\widehat{X}_y|]$ may be different for different $y$. However, $|\widehat{X}_y|$ has a high variance, because the Fourier transform is unstable to small deformations (Bruna and Mallat, 2013).

Fourier modulus descriptors can be improved by using filters $\psi$ that have a localized support in space. Theorem 1 shows that the phase of $X_y * \psi$ is also uniformly distributed in $[0, 2\pi]$. It results that $\mathbb{E}[X_y * \psi] = 0$, and $x * \psi$ still provides no information for linear classification. Applying a modulus similarly computes approximate invariants to small translations: $|x_\tau * \psi| \approx |x * \psi|$, with an error bounded by $\sigma\,|\tau|\,\|x\|_2$. More generally, these *phase collapses* compute approximate invariants to deformations which are well approximated by translations over the support of $\psi$. This representation improves linear classification by creating different non-zero class means $\mathbb{E}[|X_y * \psi|]$ while achieving a lower variance than Fourier coefficients, as it is stable to deformations (Bruna and Mallat, 2013).

Image classes are usually not invariant to global translations, because of e.g. centered subjects or the sky located in the topmost part of the image. However, classes are often invariant to local translations, up to an unknown maximum scale. This is captured by the notion of local stationarity, which means that the probability distribution of $X_y$ is nearly invariant to translations smaller than some maximum scale (Priestley, 1965). The above discussion remains valid if $X_y$ is only locally stationary over a domain larger than the support of $\psi$. The use of so-called "windowed absolute spectra" $\mathbb{E}[|X_y * \psi|]$ for locally stationary processes has previously been studied in Tygert et al. (2016).

**Real or complex networks**   The use of complex numbers is a mathematical abstraction which allows diagonalizing translations, which are then represented by complex phases. It provides a mathematical interpretation of filtering operations performed on real numbers. We show that a real network can still implement complex phase collapses.

In the first layer of a CNN, one can observe that filters are often oscillatory patterns with small supports, where some filters have nearly the same orientation and frequency but with a phase shifted by some $\alpha$ (Krizhevsky et al., 2012). We reproduce in Appendix A a figure from Shang et al. (2016) which evidences this phenomenon. It shows that real filters may be arranged in groups $(\psi_\alpha)_\alpha$ that

can be written $\psi_\alpha = \mathrm{Re}(e^{-i\alpha}\psi)$ for a single complex filter $\psi$ and several phases $\alpha$. A CNN with complex filters is thus a structured real-valued CNN, where several real filters $(\psi_\alpha)_\alpha$ have been regrouped into a single complex filter $\psi$. This structure simplifies the mathematical interpretation of non-linearities by explicitly defining the phase, which is otherwise a hidden variable relating multiple filter outputs within each layer.

A phase collapse is explicitly computed with a complex wavelet filter and a modulus. It can also be implicitly calculated by real-valued CNNs. Indeed, for any real-valued signal $x$, we have:

$$|x * \psi| = \frac{1}{2} \int_{-\pi}^{\pi} \mathrm{ReLU}(x * \psi_\alpha) \, d\alpha. \tag{4}$$

Furthermore, this integral is well approximated by a sum over 4 phases, allowing to compute complex moduli with real-valued filters and ReLUs without biases. See Appendix D for a proof of eq. (4) and its approximation.

## 3    Learned Scattering Network with Phase Collapses

This section introduces a learned scattering transform, which is a highly structured CNN architecture relying on phase collapses and reaching ResNet accuracy on the ImageNet (Russakovsky et al., 2015) and CIFAR-10 (Krizhevsky, 2009) datasets.

**Scattering transform**    Theorem 1 proves that a modulus applied to the output of a complex wavelet filter produces a locally invariant descriptor. This descriptor can then be subsampled, depending upon the filter's bandwidth. We briefly review the scattering transform (Mallat, 2012; Bruna and Mallat, 2013), which iterates phase collapses.

A scattering transform over $J$ scales is implemented with a network of depth $J$, whose filters are specified by the choice of wavelet. Let $x_0 = x$. For $0 \le j < J$, the $(j+1)$-th layer $x_{j+1}$ is computed by applying a phase collapse on the $j$-th layer $x_j$. It is implemented by a modulus which collapses the phases created by a wavelet filtering operator $W$:

$$x_{j+1} = \big|W x_j\big|. \tag{5}$$

The operator $W$ is defined with Morlet filters (Bruna and Mallat, 2013). It has one low-pass filter $g_0$, and $L$ zero-mean complex band-pass filters $(g_\ell)_\ell$, having an angular direction $\ell\pi/L$ for $0 < \ell \le L$. It thus transforms an input image $x(u)$ into $L+1$ sub-band images which are subsampled by 2:

$$Wx(u, \ell) = x * g_\ell(2u). \tag{6}$$

The cascade of $j$ low-pass filters $g_0$ with a final band-pass filter $g_\ell$, each followed by a subsampling, computes wavelet coefficients at a scale $2^j$. One can also modify the wavelet filtering $W$ to compute intermediate scales $2^{j/2}$, as explained in Appendix G. The spatial subsampling is then only computed every other layer, and the depth of the network becomes twice larger. Applying a linear classifier on such a scattering transform gives good results on simple classification problems such as MNIST (LeCun et al., 2010). However, results are well below ResNet accuracy on CIFAR-10 and ImageNet, as shown in Table 1.

**Learned Scattering**    The prior work of Zarka et al. (2021) showed that a scattering transform can reach ResNet accuracy by incorporating learned $1 \times 1$ convolutional operators and soft-thresholding non-linearities in-between wavelet filters. In contrast, we introduce a Learned Scattering architecture whose sole non-linearity is a phase collapse. It shows that neither biases nor thresholdings are necessary to reach a high accuracy in image classification. A similar result had previously been obtained on image denoising (Mohan et al., 2019).

The Learned Scattering (LScat) network inserts in eq. (5) a learned complex $1 \times 1$ convolutional operator $P_j$ which reduces the channel dimensionality of each layer $x_j$ before each phase collapse:

$$x_{j+1} = \big|W P_j x_j\big|. \tag{7}$$

Similar architectures which separate space-mixing and channel-mixing operators had previously been studied in the context of basis expansion (Qiu et al., 2018; Ulicny et al., 2019) or to filter scattering

Table 1: Error of linear classifiers applied to a scattering (Scat), learned scattering (LScat) and learned scattering with skip connections (+ skip), on CIFAR-10 and ImageNet. The last column gives the single-crop error of ResNet-20 for CIFAR-10 and ResNet-18 for ImageNet, taken from `https://pytorch.org/vision/stable/models.html`.

|  |  | Scat | LScat | LScat + skip | ResNet |
|---|---|---|---|---|---|
| **CIFAR-10** | Top-1 error (%) | 27.7 | 11.7 | 7.7 | 8.8 |
| **ImageNet** | Top-5 error (%) | 54.1 | 15.2 | 11.0 | 10.9 |
|  | Top-1 error (%) | 73.0 | 35.9 | 30.1 | 30.2 |

channels (Cotter and Kingsbury, 2019). This separation is also a major feature of recent architectures such as Vision Transformers (Dosovitskiy et al., 2021) or MLP-Mixer (Tolstikhin et al., 2021).

Each $P_j$ computes discriminative channels whose spatial variability is eliminated by the phase collapse operator. Their role is further discussed in Section 5. Table 1 gives the accuracy of a linear classifier applied to the last layer of this Learned Scattering. It provides an important improvement over a scattering transform, but it does not yet reach the accuracy of ResNet-18.

Including the linear classifier, the architecture uses a total number of layers $J + 1 = 12$ for ImageNet and $J + 1 = 9$ for CIFAR, by introducing intermediate scales. The number of channels of $P_j x_j$ is the same as in a standard ResNet architecture (He et al., 2016) and remains no larger than $512$. More details are provided in Appendix G.

**Skip-connections across moduli**  Equation (7) imposes that all phases are collapsed at each layer, after computing a wavelet transform. More flexibility is provided by adding a skip-connection which concatenates $WP_j x_j$ with its modulus:

$$x_{j+1} = \left[ \left| WP_j x_j \right|, \, WP_j x_j \right]. \tag{8}$$

The skip-connection produces a cascade of convolutional filters $W$ without non-linearities in-between. The resulting convolutional operator $WW \cdots W$ is a "wavelet packet" transform which generalizes the wavelet transform (Coifman and Wickerhauser, 1992). Wavelet packets are obtained as the cascade of low-pass and band-pass filters $(g_\ell)_\ell$, each followed by a subsampling. Besides wavelets, wavelet packets include filters having a larger spatial support and a narrower Fourier bandwidth. A wavelet packet transform is then similar to a local Fourier transform. Applying a modulus on such wavelet packet coefficients defines local spatial invariants over larger domains.

As discussed in Section 2, image classes are usually invariant to local rather than global translations. Section 2 explains that a phase collapse improves discriminability for image classes that are locally translation-invariant over the filter's support. Indeed, phases of wavelet coefficients are then uniformly distributed over $[0, 2\pi]$, yielding zero-mean coefficients for all classes. At scales where there is no local translation-invariance, these phases are no longer uniformly distributed, and they encode information about the spatial localization of features. Introducing a skip-connection provides the flexibility to choose whether to eliminate phases at different scales or to propagate them up to the last layer. Indeed, the next $1 \times 1$ operator $P_{j+1}$ linearly combines $\left| WP_j x_j \right|$ and $WP_j x_j$ and may learn to use only one of these. This adds some localization information, which appears to be important.

Table 1 shows that the skip-connection indeed improves classification accuracy. A linear classifier on this Learned Scattering reaches ResNet-18 accuracy on CIFAR-10 and ImageNet. It demonstrates that collapsing appropriate phases is sufficient to obtain a high accuracy on large-scale classification problems. Learning is reduced to $1 \times 1$ convolutions $(P_j)_j$ across channels.

## 4 Phase Collapses Versus Amplitude Reductions

We now compare phase collapses with amplitude reductions, which are non-linearities which preserve the phase and act on the amplitude. We show that the accuracy of a Learned Scattering network is considerably reduced when the phase collapse modulus is replaced by soft-thresholdings with learned biases. This result remains true for other amplitude reductions and architectures.

**Thresholding and sparsity** A complex soft-thresholding reduces the amplitude of its input $z = |z|e^{i\varphi}$ by $b$ while preserving the phase: $\rho_b(z) = \text{ReLU}(|z| - b)\,e^{i\varphi}$. Similarly to its real counterpart, it is obtained as the proximal operator of the complex modulus (Yang et al., 2012):

$$\rho_b(z) = \arg\min_{w\in\mathbb{C}}\; b|w| + \frac{1}{2}|w - z|^2. \tag{9}$$

Soft-thresholdings and moduli have opposite properties, since soft-thresholdings preserve the phase while attenuating the amplitude, whereas moduli preserve the amplitude while eliminating the phase. In contrast, ReLUs with biases are more general non-linearities which can act both on phase and amplitude. This is best illustrated over $\mathbb{R}$ where the phase is replaced by the sign, through the even-odd decomposition. If $z \in \mathbb{R}$ and $b \geq 0$, then the even part of $\text{ReLU}(z - b)$ is $\text{ReLU}(|z| - b)$, which is an absolute value with a dead-zone $[-b, b]$. When $b = 0$, it becomes an absolute value $|z|$. The odd part is a soft-thresholding $\rho_b(z) = \text{sign}(z)\,\text{ReLU}(|z| - b)$. Over $\mathbb{C}$, a similar result can be obtained through the decomposition into phase harmonics (Mallat et al., 2019).

We have explained how phase collapses can improve the classification accuracy of locally stationary processes by separating class means $\mathbb{E}\big[\big|X_y * \psi\big|\big]$. In contrast, since the phase of $X_y * \psi$ is uniformly distributed for such processes, then it is also true of $\rho_b(X_y * \psi)$. This implies that $\mathbb{E}\big[\rho_b(X_y * \psi)\big] = 0$ for all $b$. Class means of locally stationary processes are thus not separated by a thresholding.

When class means $\mathbb{E}[X_y * \psi]$ are separated, a soft-thresholding of $X_y * \psi$ may however improve classification accuracy. If $X_y * \psi$ is sparse, then a soft-thresholding $\rho_b(X_y * \psi)$ reduces the within-class variance (Donoho and Johnstone, 1994; Zarka et al., 2021). Coefficients below the threshold may be assimilated to unnecessary "clutter" which is set to 0. To improve classification, convolutional filters must then produce high-amplitude coefficients corresponding to discriminative "features".

**Phase collapses versus amplitude reductions** A Learned Scattering with phase collapses preserves the amplitudes of wavelet coefficients and eliminates their phases. On the opposite, one may use a non-linearity which preserves the phases of wavelet coefficients but attenuates their amplitudes, such as a soft-thresholding. We show that such non-linearities considerably degrade the classification accuracy compared to phase collapses.

Several previous works made the hypothesis that sparsifying neural responses with thresholdings is a major mechanism for improving classification accuracy (Sun et al., 2018; Sulam et al., 2018; 2019; Mahdizadehaghdam et al., 2019; Zarka et al., 2020; 2021). The dimensionality of sparse representations can then be reduced with random filters which implement a form of compressed sensing (Donoho, 2006; Candes et al., 2006). The interpretation of CNNs as compressed sensing machines with random filters has been studied (Giryes et al., 2015), but it never led to classification results close to e.g. ResNet accuracy.

To test this hypothesis, we replace the modulus non-linearity in the Learned Scattering architecture with thresholdings, or more general phase-preserving non-linearities. A Learned Amplitude Reduction Scattering applies a non-linearity $\rho(z)$ which preserves the phases of wavelet coefficients $z = |z|e^{i\varphi}$: $\rho(z) = e^{i\varphi}\,\rho(|z|)$. Without skip-connections, each layer $x_{j+1}$ is computed from $x_j$ by:

$$x_{j+1} = \rho(WP_j x_j), \tag{10}$$

and with skip-connections:

$$x_{j+1} = \Big[\rho(WP_j x_j),\, WP_j x_j\Big]. \tag{11}$$

A soft-thresholding is defined by $\rho(|z|) = \text{ReLU}(|z| - b)$ for some threshold $b$. We also define an amplitude hyperbolic tangent $\rho(|z|) = (e^{|z|} - e^{-|z|})/(e^{|z|} + e^{-|z|})$, an amplitude sigmoid as $\rho(|z|) = (1 + e^{-a\log|z| - b})^{-1}$ and an amplitude soft-sign as $\rho(|z|) = |z|/(1 + |z|)$. The soft-thresholding and sigmoid parameters $a$ and $b$ are learned for each layer and each channel.

We evaluate the classification performance of a Learned Amplitude Reduction Scattering on CIFAR-10, by applying a linear classifier on the last layer. Classification results are given in Table 2 for different amplitude reductions, with or without skip-connections. Learned Amplitude Reduction Scatterings yield much larger errors than a Learned Scattering with phase collapses. Without skip-connections, they are even above a scattering transform, which also uses phase collapses but does not

Table 2: Top-1 error (in %) on CIFAR-10 with a linear classifier applied to a Scattering network (Scat) and several Learned Scattering networks (LScat) with several non-linearities. They include a modulus (Mod), an amplitude soft-thresholding (Thresh), an amplitude hyperbolic tangent (ATanh), an amplitude sigmoid (ASigmoid), and an amplitude Soft-sign (ASign).

| | Scat | LScat | | | | |
| --- | --- | --- | --- | --- | --- | --- |
| | | Mod | AThresh | ATanh | ASigmoid | ASign |
| Without skip | 27.7 | 11.7 | 36.7 | 40.7 | 38.5 | 39.9 |
| With skip | - | 7.7 | 22.5 | 19.2 | 17.0 | 19.5 |

have learned $1 \times 1$ convolutional projections $(P_j)_j$. It demonstrates that high accuracies result from phase collapses without biases, as opposed to amplitude reduction operators including thresholdings, which learn bias parameters. Similar experiments in the real domain with a standard ResNet-18 architecture on the ImageNet dataset can be found in Appendix B.

**ReLUs with biases** Most CNNs, including ResNets, use ReLUs with biases. A ReLU with bias simultaneously affects the sign and the amplitude of its real input. Over complex numbers, it amounts to transforming the phase and the amplitude. These numerical experiments show that accuracy improvements result from acting on the sign or phase rather than the amplitude. Furthermore, this can be constrained to collapsing the phase of wavelet coefficients while preserving their amplitude.

Several CNN architectures have demonstrated a good classification accuracy with iterated thresholding algorithms, which increase sparsity. However, all these architecture also modified the sign of coefficients by computing *non-negative* sparse codes (Sun et al., 2018; Sulam et al., 2018; Mahdizade-haghdam et al., 2019) or with additional ReLU or modulus layers (Zarka et al., 2020; 2021). It seems that it is the sign or phase collapse of these non-linearities which is responsible for good classification accuracies, as opposed to the calculation of sparse codes through iterated amplitude reductions.

## 5 ITERATING PHASE COLLAPSES AND AMPLITUDE REDUCTIONS

We now provide a theoretical justification to the above numerical results in simplified mathematical frameworks. This section studies the behavior of phase collapses and amplitude reductions when they are iterated over several layers. It shows that phase collapses benefit from iterations over multiple layers, whereas there is no significant gain in performance when iterating amplitude reductions.

### 5.1 ITERATED PHASE COLLAPSES

We explain the role of iterated phase collapses with multiple filters at each layer. Classification accuracy is improved through the creation of additional dimensions to separate class means. The learned projectors $(P_j)_j$ are optimized for this separation.

We consider the classification of stationary processes $X_y \in \mathbb{R}^d$, corresponding to different image classes indexed by $y$. Given a realization $x$ of $X_y$, and because of stationarity, the optimal linear classifier is calculated from the empirical mean $1/d \sum_u x(u)$. It computes an optimal linear estimation of $\mathbb{E}[X_y(u)] = \mu_y$. If all classes have the same mean $\mu_y = \mu$, then all linear classifiers fail.

As explained in Section 2, linear classification can be improved by computing $(|x * \psi_k|)_k$ for some wavelet filters $(\psi_k)_k$. These phase collapses create additional directions with non-zero means which may separate the classes. If $X_y$ is stationary, then $|X_y * \psi_k|$ remains stationary for any $\psi_k$. An optimal linear classifier applied to $(|x * \psi_k(u)|)_k$ is thus obtained by a linear combination of all empirical means $(1/d \sum_u |x * \psi_k(u)|)_k$. They are proportional to the $\ell^1$ norm $\|x * \psi_k\|_1$, which is a measure of sparsity of $x * \psi_k$.

If linear classification on $(|x * \psi_k(u)|)_k$ fails, it reveals that the means $\mathbb{E}[|X_y * \psi_k(u)|] = \mu_{y,k}$ are not sufficiently different. Separation can be improved by considering the spatial variations of $|X_y * \psi_k(u)|$ for different $y$. These variations can be revealed by a phase collapse on a new set of

wavelet filters $\psi_{k'}$, which computes $(\||x * \psi_k| * \psi_{k'}|)_{k,k'}$. This phase collapse iteration is the principle used by scattering transforms to discriminate textures (Bruna and Mallat, 2013; Sifre and Mallat, 2013): each successive phase collapse creates additional directions to separate class means.

However, this may still not be sufficient to separate class means. More discriminant statistical properties may be obtained by linearly combining $(|x * \psi_k|)_k$ across $k$ before applying a new filter $\psi_{k'}$. In a Learned Scattering with phase collapse, this is done with a linear projector $P_1$ across the channel indices $k$, before computing a convolution with the next filter $\psi_{k'}$. The $1 \times 1$ operator $P_1$ is optimized to improve the linear classification accuracy. It amounts to learning weights $w_k$ such that $\mathbb{E}[|\sum_k w_k|X_y * \psi_k| * \psi_{k'}|]$ is as different as possible for different $y$. Because these are proportional to the $\ell^1$ norms $\||\sum_k w_k|x * \psi_k| * \psi_{k'}\|_1$, it means that the images $\sum_k w_k|x * \psi_k| * \psi_{k'}$ have different sparsity levels depending upon the class $y$ of $x$. The weights $(w_k)_k$ of $P_1$ can thus be interpreted as features along channels providing different sparsifications for different classes. A Learned Scattering network learns such $P_j$ at each scale $j$.

## 5.2 ITERATED AMPLITUDE REDUCTIONS

Sparse representations and amplitude reduction algorithms may improve linear classification by reducing the variance of class mean estimations, which can be interpreted as clutter removal. Such approaches are studied in Zarka et al. (2021) by modeling the clutter as an additive white noise. Although a single thresholding step may improve linear classification, we show that iterating more than one thresholding does not improve the classification accuracy, if no phase collapses are inserted.

To understand these properties, we consider the discrimination of classes $X_y$ for which class means $\mathbb{E}(X_y) = \mu_y$ are all different. If there exists $y'$ such that $\|\mu_y - \mu_{y'}\|$ is small, then the class $y$ can still be discriminated from $y'$ if we can estimate $\mathbb{E}(X_y)$ sufficiently accurately from a single realization $x$ of $X_y$. This is a mean estimation problem. Suppose that $X_y = \mu_y + \mathcal{N}(0, \sigma^2)$ is contaminated with Gaussian white noise, where the noise models some clutter. Suppose also that there exists a linear orthogonal operator $D$ such that $D\mu_y$ is sparse for every $y$, and hence has its energy concentrated in few non-zero coefficients. Such a $D$ may be computed by minimizing the expected $\ell^1$ norm $\sum_y \mathbb{E}[\|DX_y\|_1]$. The estimation of $\mu_y$ can be improved with a soft-thresholding estimator (Donoho and Johnstone, 1994), which sets to zero all coefficients below a threshold $b$ proportional to $\sigma$. It amounts to computing $\rho_b(Dx)$, where $\rho_b$ is a soft-thresholding.

However, we explain below why this approach cannot be further iterated without inserting phase collapses. The reason is that a sparse representation $\rho_b(Dx)$ concentrates its entropy in the phases of the coefficients, rather than their amplitude. We then show that such processes cannot be further sparsified, which means that a second thresholding $\rho_{b'}(D'\rho_b(Dx))$ will not reduce further the variance of class mean estimators. This entails that a model of within-class variability relying on amplitude reductions cannot be the sole mechanism behind the performance of deep networks.

Iterating amplitude reductions may however be useful if it is alternated with another non-linearity which partly or fully collapses phases. Reducing the entropy of the phases of $\rho_b(Dx)$ allows $\rho_{b'}D'$ to further sparsify the process and hence further reduce the within-class variability. As mentioned in Section 4, this is the case for previous work which used iterated sparsification operators (Sun et al., 2018; Sulam et al., 2018; Mahdizadehaghdam et al., 2019). Indeed, these networks compute non-negative sparse codes where sparsity is enforced with a ReLU, which acts both on phases and amplitudes. Our results shows that the benefit of iterating non-negative sparse coding comes from the sign collapse due to the non-negativity constraint.

We now qualitatively demonstrate these claims with two theorems. We first show that finding the sparsest representation of a random process (i.e., minimizing its $\ell^1$ norm) is the same as maximizing a lower bound on the entropy of its phases.

**Theorem 2.** *Let $X$ denote a random vector in $\mathbb{C}^d$ with a probability density $p$. Let $H(X)$ be the entropy of $X$ with respect to the Lebesgue measure:*

$$H(X) = -\int p(x) \log p(x) \, \mathrm{d}x.$$

*If $D \in U(d)$ is a unitary operator then:*

$$H\Big(\varphi(DX) \,\Big|\, |DX|\Big) \geq H(X) - d - 2d \log\left(\frac{1}{d}\mathbb{E}[\|DX\|_1]\right),$$

*where $\varphi(DX) \in [0, 2\pi]^d$ (resp. $|DX| \in \mathbb{R}_+^d$) is the random process of the entry-wise phases (resp. moduli) of $DX$.*

The proof is in Appendix E. This theorem gives a lower-bound on the conditional entropy of the phases of $DX$ with a decreasing function of the expected $\ell^1$ norm of $DX$. Minimizing over $D$ this expected $\ell^1$ norm amounts to maximizing the lower bound on $H\Big(\varphi(DX) \,\Big|\, |DX|\Big)$. An extreme situation arises when this entropy reaches its maximal value of $d \log(2\pi)$. In this case, the phase $\varphi(DX)$ has a maximum-entropy distribution and is therefore uniformly distributed in $[0, 2\pi]^d$. Moreover, in this extreme case $\varphi(DX)$ is independent from $|DX|$, since its conditional distribution does not depend on $|DX|$. Such statistical properties have previously been observed on wavelet coefficients of natural images (Rao et al., 2001), where the wavelet transform seems to be a nearly optimal sparsifying unitary dictionary.

The second theorem considers the extreme case of a random process whose phases are conditionally independent and uniform. It proves that such a process cannot be significantly sparsified with a change of basis.

**Theorem 3.** *Assume that $\varphi(\rho_b(DX))$ is uniformly distributed in $[0, 2\pi]^d$ and independent from $|\rho_b(DX)|$. Then there exists a constant $C_d > 0$ which depends on the dimension d, such that for any $D' \in U(d)$,*

$$\mathbb{E}\big[\|D'\rho_b(DX)\|_1\big] \geq C_d \mathbb{E}[\|\rho_b(DX)\|_1].$$

The proof is in Appendix F. This theorem shows that random processes with conditionally independent and uniform phases have an $\ell^1$ norm which cannot be significantly decreased by any unitary transformation. Numerical evaluations suggest that the constant $C_d$ may be chosen to be $\sqrt{\pi}/2 \approx 0.886$, independently of the dimension $d$. This constant arises as the value of $\mathbb{E}[|Z|]$ when $Z$ is a complex normal random variable with $\mathbb{E}[|Z|^2] = 1$.

These two theorems explain qualitatively that linear classification on $\rho_b(Dx)$ cannot be improved by another thresholding that would take advantage of another sparsification operator. Indeed, Theorem 2 shows that if $\rho_b(Dx)$ is sparse, then its phases have random fluctuations of high entropy. Theorem 3 indicates that such random phases prevent a further sparsification of $\rho_b(Dx)$ with some linear operator $D'$. Applying a second thresholding $\rho_{b'}(D'\rho_b(Dx))$ thus cannot significantly reduce the variance of class mean estimators.

## 6 CONCLUSION

This paper studies the improvement of linear separability for image classification in deep convolutional networks. We show that it mostly relies on a phase collapse phenomenon. Eliminating the phase of wavelet coefficients improves the separation of class means. We introduced a Learned Scattering network with wavelet phase collapses and learned $1 \times 1$ convolutional filters $(P_j)_j$, which reaches ResNet accuracy. The learned $1 \times 1$ operators $(P_j)$ enhance discriminability by computing channels that have different levels of sparsity for different classes.

When class means are separated, thresholding non-linearities can improve classification by reducing the variance of class mean estimators. When used alone, the classification performance is poor over complex datasets such as ImageNet or CIFAR-10, because class means are not sufficiently separated. Furthermore, the iteration of thresholdings on sparsification operators requires intermediary phase collapses.

These results show that linear separation of classes result from acting on the sign or phase of network coefficients rather than their amplitude. Furthermore, this can be constrained to collapsing the phase of wavelet coefficients while preserving their amplitude. The elimination of spatial variability with phase collapses is thus both necessary and sufficient to linearly separate classes on complex image datasets.

## REPRODUCIBILITY STATEMENT

The code to reproduce the experiments of the paper is available at `https://github.com/FlorentinGuth/PhaseCollapse`. All experimental details and hyperparameters are also provided in Appendix G.

## ACKNOWLEDGMENTS

This work was supported by a grant from the PRAIRIE 3IA Institute of the French ANR-19-P3IA-0001 program. We would like to thank the Scientific Computing Core at the Flatiron Institute for the use of their computing resources. We also thank Antoine Brochard, Brice Ménard and Rudy Morel for heplful comments.

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

## A  PAIRED ALEXNET FILTERS

Section 2 explains that real networks can still implement phase collapses. This is done with several real filters $\psi_\alpha = \mathrm{Re}(e^{-i\alpha}\psi)$ which correspond to several phases $\alpha$ of the same complex filter $\psi$. Shang et al. (2016) showed that the filters in e.g. the first layer of AlexNet (Krizhevsky et al., 2012) can indeed be grouped in such a way. For the sake of completeness, we reproduce in Figure 2 a figure from Shang et al. (2016). This suggests that real-valued networks may indeed implement phase collapses using eq. (4).

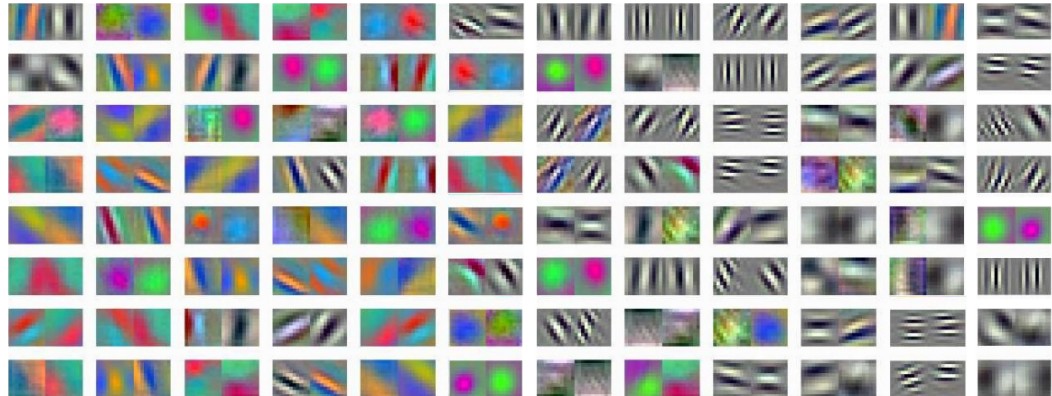

Figure 2: First-layer filters from AlexNet (Krizhevsky et al., 2012). They have been paired so that they approximately correspond to two different phases of the same complex filter $\psi$. Figure reproduced from Shang et al. (2016).

## B  PHASE COLLAPSE VERSUS AMPLITUDE REDUCTION WITH RESNET

We now evaluate the classification error of phase collapses and amplitude reduction non-linearities in the real domain. We use a standard ResNet-18 architecture without biases. We replace the ReLU non-linearity by an absolute value or sign collapse $|x|$ and several sign-preserving (i.e., odd) non-linearities. They include a soft-thresholding $\rho_b(x) = \mathrm{sign}(x)\,\mathrm{ReLU}(|x| - b)$, an hyperbolic tangent $\rho(x) = (e^x - e^{-x})/(e^x + e^{-x})$, and a soft-sign $\rho(x) = x/(1 + |x|)$. We do not report results for an amplitude sigmoid $\rho(x) = \mathrm{sign}(x)(1 + e^{-a\log|x|-b})^{-1}$ because of optimization instabilities when learning the parameters $a$ and $b$.

Classification results on the ImageNet dataset are given in Table 3. The error of bias-free ReLUs and sign collapses are comparable to a standard ResNet-18, and confirm that sign collapses are sufficient to reach such accuracies. In contrast, the performance of amplitude reduction non-linearities, which preserve the sign of network coefficients, is significantly worse. The conclusions of Section 4 thus still hold in the real domain and when the spatial filters are not constrained to be wavelets.

Table 3: Classification errors on ImageNet of bias-free ResNet-18 (BFResNet) architectures with several non-linearities. They include a ReLU, an absolute value which performs sign collapses (Abs), a soft-thresholding (Thresh), a hyperbolic tangent (Tanh), and a soft-sign (Sign). They are compared to the original ResNet-18 architecture, which uses a ReLU and learns biases.

|  | ResNet | BFResNet | | | | |
| --- | --- | --- | --- | --- | --- | --- |
|  |  | ReLU | Abs | Thresh | Tanh | Sign |
| Top-5 error (%) | 10.9 | 12.3 | 13.9 | 25.7 | 22.4 | 24.2 |
| Top-1 error (%) | 30.2 | 32.6 | 35.3 | 50.0 | 44.6 | 49.3 |

## C PROOF OF THEOREM 1

We have:

$$\|x_\tau * \psi - e^{-i\xi \cdot \tau}(x * \psi)\|_\infty = \|x * (\psi_\tau - e^{-i\xi \cdot \tau}\psi)\|_\infty \qquad \text{by covariance of convolution,}$$
$$\leq \|\psi_\tau - e^{-i\xi \cdot \tau}\psi\|_2 \|x\|_2 \qquad \text{by Young's inequality,}$$

and then:

$$\|\psi_\tau - e^{-i\xi \cdot \tau}\psi\|_2^2 = \frac{1}{(2\pi)^2} \int_{[-\pi,\pi]^2} |\widehat{\psi_\tau}(\omega) - e^{-i\xi \cdot \tau}\widehat{\psi}(\omega)|^2 \mathrm{d}\omega \qquad \text{by Plancherel,}$$
$$= \frac{1}{(2\pi)^2} \int_{[-\pi,\pi]^2} |e^{-i\omega \cdot \tau}\widehat{\psi}(\omega) - e^{-i\xi \cdot \tau}\widehat{\psi}(\omega)|^2 \mathrm{d}\omega \qquad \text{since } \psi_\tau(u) = \psi(u - \tau),$$
$$= \frac{1}{(2\pi)^2} \int_{[-\pi,\pi]^2} |e^{-i\omega \cdot \tau} - e^{-i\xi \cdot \tau}|^2 |\widehat{\psi}(\omega)|^2 \mathrm{d}\omega$$
$$\leq \frac{1}{(2\pi)^2} \int_{[-\pi,\pi]^2} |(\omega - \xi) \cdot \tau|^2 |\widehat{\psi}(\omega)|^2 \mathrm{d}\omega \qquad \text{since } x \in \mathbb{R} \mapsto e^{ix} \text{ is 1-Lipschitz,}$$
$$\leq \frac{1}{(2\pi)^2} \int_{[-\pi,\pi]^2} |\omega - \xi|^2 |\tau|^2 |\widehat{\psi}(\omega)|^2 \mathrm{d}\omega \qquad \text{by Cauchy-Schwarz,}$$
$$= \sigma^2 |\tau|^2,$$

which leads to the desired result of eq. (3):

$$\|x_\tau * \psi - e^{-i\xi \cdot \tau}(x * \psi)\|_\infty \leq \sigma |\tau| \|x\|_2.$$

## D PROOF OF EQUATION (4)

We have:
$$\mathrm{ReLU}(x * \psi_\alpha) = \mathrm{ReLU}(x * \mathrm{Re}(e^{-i\alpha}\psi)) = \mathrm{ReLU}(\mathrm{Re}(e^{-i\alpha}x * \psi)),$$

since $x$ is real. By writing: $x * \psi = |x * \psi|e^{i\varphi(x*\psi)}$ where $\varphi(x * \psi)$ is the phase of $x * \psi$, this leads to:

$$\mathrm{ReLU}(\mathrm{Re}(e^{-i\alpha}x * \psi)) = \mathrm{ReLU}(\mathrm{Re}(|x * \psi|e^{i(\varphi(x*\psi)-\alpha)}))$$
$$= \mathrm{ReLU}(|x * \psi|\cos(\varphi(x * \psi) - \alpha))$$
$$= |x * \psi| \mathrm{ReLU}(\cos(\varphi(x * \psi) - \alpha)),$$

since ReLU activation is positive-homogeneous of degree 1. Thus:

$$\frac{1}{2} \int_{-\pi}^{\pi} \mathrm{ReLU}(x * \psi_\alpha)\mathrm{d}\alpha = \frac{1}{2} \int_{-\pi}^{\pi} |x * \psi| \mathrm{ReLU}(\cos(\varphi(x * \psi) - \alpha))\mathrm{d}\alpha$$
$$= \frac{1}{2}|x * \psi| \int_{-\pi-\varphi(x*\psi)}^{\pi-\varphi(x*\psi)} \mathrm{ReLU}(\cos(-\alpha))\mathrm{d}\alpha \qquad \text{with a change of variable,}$$
$$= \frac{1}{2}|x * \psi| \int_{-\pi}^{\pi} \mathrm{ReLU}(\cos(\alpha))\mathrm{d}\alpha \qquad \text{since cos is } 2\pi \text{ periodic and even,}$$
$$= \frac{1}{2}|x * \psi| \int_{-\pi/2}^{\pi/2} \cos(\alpha)\mathrm{d}\alpha$$
$$= |x * \psi|.$$

For $z \in \mathbb{C}$, we have $|z| = \sqrt{|\mathrm{Re}(z)|^2 + |\mathrm{Im}(z)|^2} \approx |\mathrm{Re}(z)| + |\mathrm{Im}(z)|$ in the following sense:

$$\frac{1}{\sqrt{2}}(|\mathrm{Re}(z)| + |\mathrm{Im}(z)|) \leq |z| \leq |\mathrm{Re}(z)| + |\mathrm{Im}(z)|.$$

We can write:
$$|\mathrm{Re}(z)| = \mathrm{ReLU}(\mathrm{Re}(z)) + \mathrm{ReLU}(-\mathrm{Re}(z)),$$
$$|\mathrm{Im}(z)| = \mathrm{ReLU}(\mathrm{Im}(z)) + \mathrm{ReLU}(-\mathrm{Im}(z)).$$

and then, using $\mathrm{Im}(z) = \mathrm{Re}(e^{i\pi/2}z)$ and $e^{i\pi} = -1$:

$$|z| \approx \mathrm{ReLU}(\mathrm{Re}(z)) + \mathrm{ReLU}(\mathrm{Re}(e^{-i\pi}z)) + \mathrm{ReLU}(\mathrm{Re}(e^{-i\pi/2}z)) + \mathrm{ReLU}(\mathrm{Re}(e^{i\pi/2}z)).$$

Finally,

$$|x * \psi| = \frac{1}{2}\int_{-\pi}^{\pi} \mathrm{ReLU}(x * \psi_\alpha)\mathrm{d}\alpha \approx \sum_{\alpha \in \{-\pi/2, 0, \pi/2, \pi\}} \mathrm{ReLU}(\mathrm{Re}(x * \psi_\alpha)),$$

which shows that the integral can be well approximated with a sum of 4 phases $\alpha$ of the complex filter $\psi$.

## E  PROOF OF THEOREM 2

We first use the chain rule for the entropy:

$$H\Big(\varphi(DX)\,\Big|\,|DX|\Big) = H(|DX|, \varphi(DX)) - H(|DX|).$$

The first term is rewritten with a change of variable:

$$H(|DX|, \varphi(DX)) = H(DX) - \sum_{k=1}^{d} \mathbb{E}[\log|(DX)_k|]$$

$$= H(X) - \sum_{k=1}^{d} \mathbb{E}[\log|(DX)_k|] \qquad \text{as } D \text{ is unitary and hence } |\det(D)| = 1,$$

$$\geq H(X) - d\mathbb{E}\left[\log\left(\frac{1}{d}\|DX\|_1\right)\right] \qquad \text{by concavity},$$

$$\geq H(X) - d\log\left(\frac{1}{d}\mathbb{E}[\|DX\|_1]\right) \qquad \text{by concavity}.$$

The second term is bounded using the fact that the exponential distribution $\mathcal{E}(\lambda)$ is the maximum-entropy distribution on $\mathbb{R}_+$ with mean $\frac{1}{\lambda}$:

$$H(|DX|) \leq \sum_{k=1}^{d} H(|(DX)_k|)$$

$$\leq \sum_{k=1}^{d} \log(e\mathbb{E}[|(DX)_k|])$$

$$\leq d\log\left(\frac{e}{d}\mathbb{E}[\|DX\|_1]\right) \qquad \text{by concavity}.$$

Combining both inequalities and rearranging terms yields the stated bound:

$$H\Big(\varphi(DX)\,\Big|\,|DX|\Big) \geq H(X) - d - 2d\log\left(\frac{1}{d}\mathbb{E}[\|DX\|_1]\right).$$

## F  PROOF OF THEOREM 3

We begin with the following lemma:

**Lemma 1.** *Let $(\theta_1, \ldots, \theta_d)$ be i.i.d. uniform random variables in $[0, 2\pi]$. Then there exists a constant $C_d > 0$ such that for all $(\rho_1, \ldots, \rho_d) \in \mathbb{R}^d$, then:*

$$\mathbb{E}\left[|\sum_{k=1}^{d} \rho_k e^{i\theta_k}|\right] \geq C_d\sqrt{\sum_{k=1}^{d} \rho_k^2}.$$

This is proved by observing that the left-hand side is a norm on $\mathbb{R}^d$. One can indeed verify that it is positive definite, homogeneous and satisfies the triangle inequality. Since all norms on $\mathbb{R}^d$ are equivalent, there exists a constant $C_d > 0$ such that:

$$\mathbb{E}\left[|\sum_{k=1}^{d} \rho_k e^{i\theta_k}|\right] \geq C_d \sqrt{\sum_{k=1}^{d} \rho_k^2}.$$

for all $(\rho_1, \ldots, \rho_d) \in \mathbb{R}^d$.

Going back to the proof of Theorem 3, and letting $X' = \rho_b(DX)$, we then have:

$$\begin{aligned}
\mathbb{E}\left[\|D'X'\|_1 \,\Big|\, |X'|\right] &= \sum_{m=1}^{d} \mathbb{E}\left[|\sum_{k=1}^{d} D'_{m,k}X'_k| \,\Big|\, |X'|\right] \\
&\geq C_d \sum_{m=1}^{d} \sqrt{\sum_{k=1}^{d} |D'_{m,k}|^2 |X'_k|^2} \qquad \text{by the above lemma,} \\
&\geq C_d \sum_{m=1}^{d} \sum_{k=1}^{d} |D'_{m,k}|^2 |X'_k| \qquad \text{by concavity, because } \sum_{k=1}^{d} |D'_{m,k}|^2 = 1, \\
&= C_d \|X'\|_1 \qquad \text{because } \sum_{m=1}^{d} |D'_{m,k}|^2 = 1.
\end{aligned}$$

Taking the expectation finishes the proof:

$$\mathbb{E}\left[\|D'X'\|_1\right] \geq C_d \mathbb{E}\left[\|X'\|_1\right]. \tag{12}$$

## G  EXPERIMENTAL DETAILS

**Channel operators**    In all experiments we set $P_0 = \mathrm{Id}$, and factorize the classifier with an additional complex $1 \times 1$ convolutional operator $P_J$, which reduces the dimension before all channels and positions are linearly combined. The architectures implemented are thus also written as $\prod_{j=1}^{J} P_j \rho W$, where $\rho$ is the non-linearity. Each operator $(P_j)_{1 \leq j \leq J}$ is preceded by a standardization. It sets the complex mean $\mu = \mathbb{E}[z]$ of every channel to zero, and the real variance $\sigma^2 = \mathbb{E}[|z|^2]$ of every channel to one. This is similar to a complex 2D batch-normalization layer (Ioffe and Szegedy, 2015), but without learned affine parameters. Each operator $(P_j)_{1 \leq j \leq J}$ is additionally followed by a spatial divisive normalization (Rao et al., 2001), similarly to the local response normalization of Krizhevsky et al. (2012). It sets the norm across channels of each spatial position to one. The sizes of the $(P_j)_j$ are specified in Table 4.

The total numbers of parameters for each architecture are specified in Table 5. Learned Scattering with phase collapse have a large number of parameters compared to ResNet, despite the comparable width. This is because the predefined wavelet operator $W$ expands the dimension by a factor of $L + 1$, which means that the input dimension of the learned $(P_j)_j$ is higher than in ResNet. The skip-connection further increases this input dimension by a factor of 2.

Table 4: Number $c_j$ of complex output channels of $P_j$, $1 \leq j \leq J$. The total number of projectors is $J = 8$ for CIFAR and $J = 11$ for ImageNet.

| | $j$ | 1 | 2 | 3 | 4 | 5 | 6 | 7 | 8 | 9 | 10 | 11 |
|---|---|---|---|---|---|---|---|---|---|---|---|---|
| **CIFAR-10** | $c_j$ | 64 | 128 | 256 | 512 | 512 | 512 | 512 | 512 | - | - | - |
| **ImageNet** | $c_j$ | 32 | 64 | 64 | 128 | 256 | 512 | 512 | 512 | 512 | 512 | 256 |

Table 5: Number of real parameters (in millions) of Learned Scattering network architectures. A complex parameter is counted as two real parameters.

|  | PCScat | PCScat + skip | ResNet |
|---|---|---|---|
| **CIFAR-10** | 41.6 | 83.1 | 0.27 |
| **ImageNet** | 36.0 | 62.8 | 11.7 |

**Spatial filters**    We use elongated Morlet filters for the $L$ complex band-pass filters $(g_\ell)_\ell$ which are rotated versions of a mother wavelet $g$: $g_\ell(u) = g(r_{-\pi\ell/L}u)$, with $r_\theta$ the rotation by angle $\theta$. The mother wavelet $g$ is defined as:

$$g(u) = \frac{\sigma^2}{2\pi/s^2}(e^{i\xi\cdot u} - K)e^{-u\cdot\Sigma u/2} \qquad \text{with } \Sigma = \begin{pmatrix} \sigma^2 & 0 \\ 0 & \sigma^2 s^2 \end{pmatrix}, \tag{13}$$

Its parameters are its center frequency $\xi = ((3\pi/4)/2^\gamma, 0)$, its bandwidth $\sigma = 1.25 \times 2^{-\gamma}$, and its slant $s = 0.5$, where $2^\gamma$ designates the scale of the band-pass filter and is to be adjusted.

$g$ is rotated along $L = 8$ angles for Imagenet and $L = 4$ angles for CIFAR: $\theta_\ell = (\pi\ell/L)_{1\leq\ell\leq L}$. The $(g_\ell)_\ell$ are then discretized for numerical computations, and $K$ is adjusted so that they have a zero mean.

Finally, we use for the low frequency $g_0$ a Gaussian window:

$$g_0(u) = \frac{\sigma^2}{2\pi}e^{-\sigma^2\|u\|_2^2/2}.$$

The filters are implemented with the *Kymatio* package (Andreux et al., 2020).

Intermediate scales $2^{j/2}$ are obtained by applying a subsampling by 2 after each block of 2 layers. This introduces intermediate scales and generates a wavelet filterbank with 2 scales per octave: the filters are designed so that when $j$ low-pass filters and one band-pass filter are cascaded, with a subsampling every 2 layers, the scale of the resulting wavelet is $2^{j/2}$.

Each block comprises in its first layer a low-frequency filter $g_0^1$ with $\gamma = -1/2$ and band-pass filters with $\gamma = 0$. In the second layer, we use the same low-frequency filter $g_0^2 = g_0^1$ with $\gamma = -1/2$. The band-pass filters $g_\ell^2$ are obtained with parameters $\xi' = (\pi/\sqrt{2}, 0)$, $\sigma' = 1.25\sqrt{2/3}$, and $s' = \sqrt{0.2}$.

For CIFAR experiments, the $J = 8$ layers are grouped in 4 successive blocks of 2 layers. For ImageNet experiments, the first layer consists of band-pass elongated Morlet filters $g_\ell$ and a low-pass Gaussian window $g_0$ with $\gamma = 0$, followed by a subsampling of 2. The 10 following layers are grouped in 5 blocks of 2 layers.

**Optimization**    We use the optimizer SGD with an initial learning rate of $0.01$, a momentum of $0.9$, a weight decay of $0.0001$, and a batch size of 128. The classifier is preceded by a 2D batch-normalization layer. We use traditional data augmentation: horizontal flips and random crops for CIFAR, random resized crops of size 224 and horizontal flips for ImageNet. Classification error on ImageNet validation set is computed on a single center crop of size 224. On CIFAR, training lasts for 300 epochs and the learning rate is divided by 10 every 70 epochs. On ImageNet, training lasts for 150 epochs and the learning rate is divided by 10 every 45 epochs. All experiments ran during the preparation of this paper, including preliminary ones, required around 10k 32GB NVIDIA V100 GPU-hours.

