# OpenReview forum: "Phase Collapse in Neural Networks"
_ICLR.cc/2022/Conference — ICLR 2022 Poster_

### Official Review · Reviewer_oXQJ · 2021-11-01

**Correctness:** 4
**Technical Novelty And Significance:** 3
**Empirical Novelty And Significance:** 3
**Recommendation:** 6
**Confidence:** 2

**Main Review:**

I think the paper addresses some interesting questions and it is somewhat interesting that a network with a fixed wavelet basis can achieve reasonable performance on ImageNet classification. I also think the paper provides quite a bit of theory to support some of the claims.

However I feel the paper suffers from several weaknesses:

* It is not clear to what is the purpose of the proposed model? if it is to show that such a model can indeed be comparable to current architectures then OK, but it does not go beyond that. I don't feel there is a lot "actionable" conclusions here nor that we learned anything fundamental about how these models work.

* There is almost no discussion as to what the learnable operators P do - what happens if they are not used? how do they relate to other parts of the model and, more importantly, to what ResNets learn?

* Why are the resulting networks so parameter heavy? they have many more parameters when compared to the corresponding ResNets and perform worse. What can be done?

* How does the proposed model scale? with the number of layers? number of activations at each layer? etc.

**Summary Of The Paper:**

This paper proposes that using phase modulus operators instead of thresholding non-linearites is beneficial to classification performance in the case of scattering like networks. The authors show that such a network with some learnable operators can come close to performance of small ResNets when they use the modulo operators and performance degrades when using other non-linearieis. This is demonstrated on ImageNet and CIFAR-10 and is accompanied with relevant analysis and theory.

**Summary Of The Review:**

I think this is a potentially interesting paper but I feel it leaves quite a bit to be desired.

post author response update:

I have read the other reviewers comments as well as the author responses - it would seem that I have been harsh in my scoring and this has been revised. I think this is a valuable contribution to our understanding of the field and a nice demonstration that fixed wavelet filters can, under the right circumstances, perform as well as learned networks. My confidence in the score still remains low as I feel was not the right person to review this.

---

> ### Author Response · Authors · 2021-11-21
> **Answer to Reviewer oXQJ**
>
> We thank the reviewer for their time and comments that helped us improve the paper. We clarify below some misunderstandings. We also modified the paper to clarify these issues.
> - The purpose of the current model is to explain the role of the non-linearity in relation with the filters that are used, not to improve the algorithmic state-of-the-art. We show that a complex modulus non-linearity with fixed wavelet filters is sufficient to reach ResNet accuracy. This result is surprising because we do not use biases and the universal approximation theorem therefore does not hold, and because we do not learn the spatial filters. This result is also surprising because a large part of the research community believes that sparsity and biases are important to explain the accuracy of deep networks. We demonstrate that this is not the case by showing that modulus non-linearities, which do not enforce sparsity, provide much better results than thresholding nonlinearities that enforce sparsity. These are significant and novel contributions. We have clarified this in the paper to avoid misunderstandings.
> - We will further clarify the role of the learnable parameters $P$. They compress the number of channels which would otherwise grow exponentially. Moreover, they learn discriminative combinations of channels which considerably improve the classification accuracy. If they were not used, then there would be hundreds of billions of channels, even with 4 angles in the wavelet filterbank. This is why the scattering transform uses predefined $P_j$ which eliminates coefficients having more than two non-linearities. By comparing results with operators which are learned and not learned, we show that the classification error is reduced by at least a factor of $3$. Understanding the mathematical nature of what is learnt by the $P_j$ is an open and very difficult question, which would require having a full mathematical model of deep neural networks. Nobody is there yet.
> - The goal of the paper was not to minimize the number of parameters of the architecture, and the reviewer is right that it includes many more parameters than a ResNet. The goal is rather to constrain the architecture in order to understand the role of the non-linearities and spatial filters, without ambiguity. As the reviewer mentions, the architecture can be optimized in terms of its parameters, or scaled to increase its performance. However, reaching a higher performance with a larger network or fewer parameters would not improve the main results concerning the role of non-linearities and spatial filters. Most current research papers are devoted to optimizing performances on benchmarks to reach SOTA results. This paper intends to demonstrate some of the mathematical mechanisms which lead to good classification results with deep neural networks.

---

### Official Review · Reviewer_GfRM · 2021-11-02

**Correctness:** 3
**Technical Novelty And Significance:** 3
**Empirical Novelty And Significance:** 3
**Recommendation:** 6
**Confidence:** 4

**Main Review:**

Strengths
-The paper is interesting in that it presents theoretical results followed by solid experimental results related to the theoretical results demonstrating that the phase of the signal can be removed while improving the classification performance of DNNs
-The paper is for the most part well written
-The authors present good results on CIFAR10 and Imagenet. While the results do not seem state of the art I think they are sufficiently good in support of the authors' arguments

Weaknesses:
-Figure 1:The authors assume color printers are used (orange/blue). I suggest they use bold/solid, dashed lines to distinguish between the different steps
-Page 2: clarify why both directions (necessary & sufficient) hold. I see how the sufficient condition holds but it is not clear how the necessary direction emerges. Does it make sense to phrase this as a conjecture?
-Eq 4: It may help many readers to derive this in the appendix, it should be a quick derivation
-The ResNet results while good are not state of the art. If SOA feasible using the PCS framework? If yes why wasn't a SOA presented in the paper. If not, what do the authors believe hinders achieving SOA?
-Page 3: Clarify better when the random vector X_y represents. Does it represent the features used by the classifier? In that case I do not see how the classes are linearly separable if E[X_y] are all different. Do you mean this is a sufficient or necessary condition? For example if X_1, X_2 are 2D vectors, each representing two concentric circles. The two sets of features represent different classes but the means are the same (the circle center). Obviously we can have E[X_y] with the same mean but represent different classes. Limiting ourselves to linear separability makes the problem trivial. I suggest the author rewrite this section to clarify what they mean
Page 3: The authors make a lot of assumptions that X_y are stationary. How important is this assumption in their results? Do they think perhaps that a significant set of problems, such as those often dealt with by RNNs, are not well modelled as stationary?

**Summary Of The Paper:**

The authors present some theoretical results, followed by some experiments, in support of their argument that the linear separability for image classification in deep convolutional neural networks mostly relies on a phase collapse phenomenon. By eliminating the phase of zero mean filters they improve the separation of class means.They present a Phase Collapse Scattering network and demonstrate Resnet-like accuracy. The authors argue that phase collapse are both necessary and sufficient to discriminate class means on complex datasets.

**Summary Of The Review:**

Overall I found this an interesting paper. It is well written, and has a fairly solid experimental section in support of some not so well known theoretical ideas as to why deep neural nets achieve good classification
Overall here is my summarized assessment:

originality: The authors present some theoretical results to justify their experimental framework. The paper is not just blindly attempting different DNN architectures. From this point of view I consider this a fairly original piece of work

quality: I believe the exposition and experimental results are of sufficient quality for ICLR. The paper is in general well written, though as I indicate elsewhere certain parts need to be improved.

clarity: I mentioned above in the "Weaknesses" section some clarity issues that I noticed. The authors should address these issues. I was ambivalent as to whether the paper should be rejected because of these clarity issues, but since they can all be addressed I think I decided to be a bit lenient. But for the final ICLR submission they should be addressed.

significance: Unless I missed it, are the authors providing source code? The paper's significance and clarity will be improved if source code is provided, to make the results reproducible

---

> ### Author Response · Authors · 2021-11-21
> **Answer to Reviewer GfRM**
>
> We thank the reviewer for their time and comments that helped us improve the paper. We modified the paper according to your remarks:
> - We modified the figure for black-and-white printing according to your suggestions.
> - The formulation that “phase collapses are necessary and sufficient” is, as you rightly pointed out, a condensed statement, which we clarified in the updated version of the paper. We show in the paper that it is necessary to act on phase, with the poor numerical performance of phase-preserving non-linearities in Table 2, and with a theoretical argument on the entropy of phases of network coefficients. We also show that acting on phases through phase collapses alone is sufficient to reach ResNet-18 accuracy.
> - We have added the derivation of Equation (4) in Appendix A, and we explain how the integral can be approximated with four phases.
> - Please note that achieving SOTA is not a goal of this paper. The main contribution is to explain the role of non-linearities and phase collapses, to reach high classification accuracy in CNN architectures such as ResNets.
> - The random vector $X_y$ represents the random process whose realizations are the images of class $y$, which has been clarified. In the paragraph you mention, we explain why linear classification is known to be so poor when applied directly to raw images. For this purpose, we explain that linear classification is able to linearly discriminate classes if two necessary and sufficient conditions are met: (i) classes have different means, and (ii) intra-class variance is sufficiently small so that classes do not overlap. This second condition was mentioned in the paper as imposing that class means are sufficiently well estimated with a linear operator: this has been clarified by integrating more explicitly condition (ii) on intra-class variance. Raw images are not linearly separable because all non-zero frequencies have a zero mean. This is due to phase shifts resulting from translations. This explains why a phase collapse can improve linear classification accuracy.
> - Stationarity is not a necessary assumption for phase collapses to be useful. However, it provides a simpler mathematical framework to illustrate the underlying phenomena. Indeed, phase variations are produced by translations which are equally probable in a model of stationary classes. This is a strong constraint. Images in CIFAR or ImageNet are typically not stationary, since important structures are rather at the center. However, all structures can be translated and deformed which produce phase fluctuations as in a stationary process. This has been clarified in the paper.
> - The source code was provided as a supplementary material. We have added a reproducibility statement.

---

> > ### Comment · Reviewer_GfRM · 2021-11-28
> > **Response**
> >
> > Thank you for your response. The authors have clarified my concerns. Therefore I am leaving the score unchanged as marginally accept.

---

### Official Review · Reviewer_3Vif · 2021-11-03

**Correctness:** 3
**Technical Novelty And Significance:** 4
**Empirical Novelty And Significance:** 4
**Recommendation:** 8
**Confidence:** 4

**Main Review:**

The strengths of the paper are:
- I find the novelty and the results, both theoretical and empirical, fascinating and thought provoking. Making a link between operators like ReLUs to complex numbers, their phases, and collapse is a great idea. In particular, I find the results in Table 2 very convincing.

- The theoretical analysis is nice, and not to hard to follow, once one commits. It would be interesting to make more explicit the analysis for different types of nonlinearities. For instance, if sigmoid are soft-thresholding functions, I suppose the same is also for tanh. However, tanh tends to work better. Why is that the case, under this framework? What about other non-linear activations, like the swish function?

- The results are generally strong and quite convincing, at least in ablation comparisons.

The weaknesses of the paper:
- I found the writing involved, if not subpar for the quality of the paper. It is clear that the authors have a very good understanding of the area, however, they do not make it very easy for the average reader to understand their thought process, The sentences are not really immediately connected, that is there exist jumps, which sometimes can be inferred based on follow-up text, while other times they are simply hard to understand. It could be that for readers who are better versed in the literature, the text is easier. However, any paper should do a minimum effort to communicate the message clearly. Certainly, the introduction contains several forward references and that at least can be improved.

- I found section 2 particularly hard to follow, as various concepts are being introduced at random points, without any clear structure. Almost ironically, I couldn’t find any direct definition of what comprises phase collapse, which made matters all that much harder. While it is sort of expected/intuitive what phase collapse could mean, given that the paper is on the theoretical/conceptual side of things, it makes it hard to follow the thought process.

The only ‘definition’ I found was that phase collapse eliminates the phase of a complex number with a modules just below eq (3). It is not clear if this a general definition, just a logical deduction based on fundamental concepts, or something that the paper introduces. In any case, it is also unclear why this holds. I assume it is because the modulus of the exponential is 1, so the phase that is introduced by the exponential is eliminated?

- It is also unclear how equation (5) is derived. Again, is it a logical deduction based on the nature of the ReLU? Or is it derived based on the aforementioned definitions of filtering with wavelets and complex numbers? It should be immediately clear from the text what is expected to be background knowledge, what is derived knowledge, what is a simple deduction.

- From a technical point of view, what I found unclear was what is the point of stacking ‘phase collapse’ operators, since one of them is supposed to remove all the phases from the feature, that is, it learns translation invariant features. From deep networks, I assume that deeper layers combined the wavelet activations from the previous layers. However, I find hard to reconcile this with ‘translation invariance’. For instance, a full connected layer is translation invariant, while a convolutional layer is translation equivariant. Does phase collapse then correspond to fully connected layers, and if yes, what is the point of stacking them? After all, with fully connected layers it is also of little added value to stack them, usually.

- Complementing on the previous point, can it be that adding the skip connections is beneficial such that to make the operations also translation equivariant, thus not completely eliminate all phases and translation information? Something like that is analyzed in the bottom of page 5, but maybe this point can be furthered.

- I find the motivation of the projection operators P a logical one in the context of deep networks, however, not really convincing in the context of phase collapse scattering networks. Do they contribute anything to the amplitude or the phase? What would be the benefit, considering that all phases are anyways immediately eliminated afterward? Do they help with having class means that are well separated perhaps? Maybe there can be a motivation that relates more to the main story of the paper.

- Last, I would like to clarify that in my attempt to find out more about ‘phase collapse’ as a concept, I found out the arXiv version of the paper accidentally. I got a bit annoyed by that, as I believe it biases my opinion, and I do not find it fair against other papers.

**Summary Of The Paper:**

This is a very interesting paper that is based on scattering networks, for which it shows that the so called ‘phase collapse’ leads to state-of-the-art results on par with modern architectures, like ResNet. To derive this, the paper shows that having neural networks on complex numbers is similar to a structure deep network (like with wavelet filters) on the reals. Phase collapse is then when there is an operation, like the modulus (absolute value function), which eliminates the phase from the complex number and maintains only the amplitude. All in all, the paper shows that maintaining the amplitude while eliminating the phase, is what brings the high accuracies, while the reverse (keeping the phase, eliminating the amplitude) yields terrible accuracies. As a corollary, I would say that the paper explains why non-linearities, like ReLUs, are so successful.

**Summary Of The Review:**

Based on the novelty and the strong results, I vote for acceptance. I have one remark that I couldn’t completely understand and it would be nice if the authors could help me there. All in all, I find that I learned something from this paper, which I think is a great result for any publication.

---

> ### Author Response · Authors · 2021-11-21
> **Answer to Reviewer 3Vif**
>
> We thank the reviewer for their time and comments that helped us improve the paper.
> - We modified the paper according to your remarks: we clarified the introduction and section 2 to make them more accessible, and tried to improve the style.
> - We added the explicit definition of “phase collapse” as the removal of the complex phases of network coefficients, implemented with a complex modulus. It is motivated by eq. (3). A phase collapse over wavelet coefficients defines approximate invariants to small translations.
> - We have added some background to the definition of soft-thresholding on complex variables (eq. (5)). Similarly to real variables, it is shown to be the proximal operator of the $\ell^1$ norm.
> - A phase collapse over wavelet coefficients is not equivalent to a fully connected layer. We tried to clarify the importance of stacking phase collapses: they need to be applied at all scales because after removing its phase, a wavelet descriptor is approximately translation invariant on a limited domain, whose size is equal to the size of the wavelet’s receptive field. Moreover, it is necessary to compute wavelet coefficients at all scales to avoid losing information when calculating the phase collapse.
> - The reviewer is right that adding a skip-connection provides a complementary translation-covariant information, which provides information about spatial location. Images are not stationary processes and important structures may be at the center. Preserving information on spatial location with the skip-connection appears to be important for classification.
> - The reviewer is right that the projection operators $P_j$ ensure that class means are well separated after the phase collapse $|W|$. The phases that are collapsed are indeed those created by the joint operator $WP_j$ and thus depend on $P_j$. A more in-depth analysis of the role of these projectors is indeed an important question that we began to investigate.
> - We regret the inconvenience caused by the arXiv version of the paper, which is allowed by ICLR rules.
> - Concerning other non-linearities, tanh is indeed an amplitude reduction non-linearity as it preserves the sign. The differences between tanh and sigmoid or swish and ReLU may come from different optimization behaviors rather than significant changes in expressivity.

---

### Official Review · Reviewer_LHxR · 2021-11-07

**Correctness:** 3
**Technical Novelty And Significance:** 2
**Empirical Novelty And Significance:** 2
**Recommendation:** 8
**Confidence:** 3

**Main Review:**

### Strength:
This paper is a nice extension to the related works [Zarka et al (2020, 2021), Ulicny et al.(2019)]. The main contribution (and difference to the literature) is that they hypothesize classification performance mostly results from iterated phase collapses. They supported their hypothesis  (necessary and sufficient) by first explaining the performance of iterated phase collapses by showing that it progressively improves linear discriminability. Secondly, they replaced the phase collapse with a soft-thresholding and showed that this considerably decreases the classification accuracy.


### Correctness:
The core assumptions seem correct to me, e.g. the images classes are often locally invariant to translation. Small translations are used to be approximated by a phase shift. The empirical results are correct, the theoretical results seem reasonable, but I did not fully examine the proofs in the appendix or reproducibility of the code in the supplementary materials.

### Weakness:
Generally, the writing quality is not good and hard to read, e.g. long sentences. Page 3, before the (Krizhevsky et al., 2012) reference, I am not sure if filters are usually localized oscillatory patterns or filters usually localize oscillatory patterns? Sometimes the text is not scientifically addressed, e.g. page 4: $\mathbf{E}[X_y * \psi] = 0$ then $\mathbf{E}[\rho_b(X_y * \psi)] $ is "usually" close to zero. What does usually mean here? Formula4: "One can verify..."? It is now clear to me how. And "This integral is already well approximated by a sum over 4 phases"?  MNIST probably needs to be cited for "The MNIST database of handwritten digits (1998)". The operator "$*$" is not defined. It is also not clear to me why the authors used skip-connection-based neural networks? What about CNNs without skip connections?



>update after rebuttal: I would like to thank the reviewers for addressing my concerns. However, I found it difficult to find the changes in the rebuttal version. I now see the Results with/without skip connections in Table1 and Table2. The text is still hard to read and follow (this is also addressed by other reviewers). As promised by the authors the text would be improved (e.g. shorter sentences). In conclusion, I would increase my score.

**Summary Of The Paper:**

This paper studies within-class variability which reduces along the layers of deep neural networks. They mainly question the effect of sparsity and soft-thresholding introduced by ReLU. They show that these classification improvements by eliminating spatial within-class variabilities rather come from a phase collapse, which eliminates the phase of network coefficients. Eliminating the phase of zero-mean filters improves the separation of class means, hence increase in classification accuracy. They introduced a complex-valued neural network in which spatial filters are defined as complex multiscale wavelets and learning is reduced to $1 \times 1$ complex filters across channels. Their results show that such a network is able to reach ResNet-18 performance on CIFAR10 and ImageNet.






**Summary Of The Review:**

The paper mainly questions the effect of sparsity and soft-thresholding introduced by ReLU. The authors show that the classification improvements by eliminating spatial within-class variabilities come from a phase collapse. This counts as a contribution compared to related works. The hypothesis is clearly defined and supported using theory (and assumptions) and experiments. However, there are pitfalls that need to be addressed.

---

> ### Author Response · Authors · 2021-11-21
> **Answer to Reviewer LHxR**
>
> We thank the reviewer for their time and comments that helped us improve the paper. We modified the paper according to your remarks and to improve the style:
> - We indeed meant that the filters are localized oscillatory patterns and clarified it.
> - We clarified our statement in the updated version of the paper by explaining that when $X \ast \psi$ has a symmetric distribution around $0$ then $\mathbb E[\rho_b(X \ast \psi)] = 0$, because the thresholding preserves the phase and hence this symmetry. This particular case happens often with zero-average filters.
> - We have added the proof of eq. (4) in the appendix.
> - We already cite the MNIST webpage [Y. LeCun,  C. Cortes,  and C.J. Burges.   Mnist handwritten digit database. ATT Labs [Online]. Available: http://yann.lecun.com/exdb/mnist, 2, 2010].
> - $\ast$ denotes the convolution operator and this has been specified.
> - Concerning the use of skip-connections, to show their impact, we also give results without using them. Our results apply to networks with or without skip-connections, which is better explained.

---

> > ### Comment · Reviewer_LHxR · 2021-11-26
> > **Final phase comment**
> >
> > I would like to thank the reviewers for addressing my concerns. However again I found it difficult to find the changes in the rebuttal version e.g. results with/without skip connections. Highly recommend that you mention figures/sections when addressing concerns and questions. The text is still hard to read and follow (this is also addressed by other reviewers). I hope that the authors would consider this for the final version. I conclusion I would be happy to increase my score to acceptance (7).

---

> > > ### Author Response · Authors · 2021-11-26
> > > **Thank you for your answer**
> > >
> > > We did a first pass of rewriting and we intend to indeed further improve the writing style for the final version. Any pointer to paragraphs or sections that you think should be particularly improved would be very helpful for us. Regarding results with or without skip-connections, we would like to point out that they are mentioned in both Table 1 (columns LScat/LScat + skip) and Table 2 (rows Without/With skip).

---

### Public Comment · ~Bilal_Alsallakh1 · 2021-11-17
**Insightful work**

I want to commend the authors for their insightful work and findings.
The authors might find the work by [Tygert et al](
https://direct.mit.edu/neco/article-abstract/28/5/815/8157/A-Mathematical-Motivation-for-Complex-Valued) (Neural Computing, 2016) useful for their formalism (a preprint is available [on arxiv](https://arxiv.org/pdf/1503.03438.pdf)).

I am curious about the impact of convolution arithmetic on phase collapses (e.g. maxpooling vs BlurPool, zero padding vs circular padding), and whether non-convolutional image classifiers (e.g. vision transformers) would exhibit a similar behavior.

Keep the good work up!

---

> ### Author Response · Authors · 2021-11-21
> **Thank you for your comment**
>
> Thank you very much for your comment and your reference. The work and formalism of Tygert et al. is indeed related to ours since it uses a complex valued network, but it demonstrates different properties. We are adding this reference in the revised version of the paper.
>
> Regarding your remark on the pooling and convolution arithmetic, we have not tested the effect of maxpooling, which is more difficult to analyze mathematically because it is not a regular operator, as opposed to a BlurPool. We do not expect that it would change the results much, because part of the translation invariance is already obtained by the phase collapse. Zero-padding versus circular padding have boundary effects. To simplify mathematics, circular padding is often used, but it does not improve numerical performances compared to a zero-padding, because images are not periodic.
>
> The principle of phase collapse can apply to transformers if they learn oscillatory filters with coefficients having alternating signs. Positional embeddings allow the attention maps to mimic localized wavelet filters, and the tokenwise MLPs can compute complex moduli as linear combinations of ReLUs. In the simpler setting of MLP-Mixer (https://arxiv.org/abs/2105.01601), whose architecture is inspired from Transformers, the authors show that spatial filters of first layers are wavelets of different scales, orientations and phases as in a more traditional CNN. This shows that phase collapses may be computed by the learned network even when the architecture has no built-in prior on translation-equivariance.

---

### Decision · Program_Chairs · 2022-01-20

**Decision:**

Accept (Poster)

**Comment:**

This paper proposes that the superior performance of modern convolutional networks is partly due to a phase collapse mechanism that eliminates spatial variability while ensuring linear class separation. To support their hypothesis, authors introduce a complex-valued convolutional network (called  Learned Scattering network) which includes a phase collapse on the output of its wavelet filters and show that such network has comparable performance to ResNets but its performance degrades if the phase collapse is replaced by a threshold operator.

Reviewers are all in agreement about the novelty and significance of the work. They also find the empirical results compelling. The main weakness of this work which was highlighted by all reviewers is clarity. The paper can be significantly improved in terms of the writing. While I am recommending acceptance, I strongly recommend authors to take reviewers' feedback into account and improve the writing significantly for the final version so that more people would benefit from this paper and build on it in the future.